# Research Progress on New Environmentally Friendly Antifouling Coatings in Marine Settings: A Review

**DOI:** 10.3390/biomimetics8020200

**Published:** 2023-05-13

**Authors:** De Liu, Haobo Shu, Jiangwei Zhou, Xiuqin Bai, Pan Cao

**Affiliations:** 1School of Mechanical Engineering, Yangzhou University, Yangzhou 225127, China; 2School of International Education, Wuhan University of Technology, Wuhan 430070, China; 3State Key Laboratory of Maritime Technology and Safety, Wuhan University of Technology, Wuhan 430063, China

**Keywords:** marine biofouling, antifouling coatings, environmentally friendly, biomimetic strategy, antimicrobial peptides

## Abstract

Any equipment submerged in the ocean will have its surface attacked by fouling organisms, which can cause serious damage. Traditional antifouling coatings contain heavy metal ions, which also have a detrimental effect on the marine ecological environment and cannot fulfill the needs of practical applications. As the awareness of environmental protection is increasing, new environmentally friendly and broad-spectrum antifouling coatings have become the current research hotspot in the field of marine antifouling. This review briefly outlines the formation process of biofouling and the fouling mechanism. Then, it describes the research progress of new environmentally friendly antifouling coatings in recent years, including fouling release antifouling coatings, photocatalytic antifouling coatings and natural antifouling agents derived from biomimetic strategies, micro/nanostructured antifouling materials and hydrogel antifouling coatings. Highlights include the mechanism of action of antimicrobial peptides and the means of preparation of modified surfaces. This category of antifouling materials has broad-spectrum antimicrobial activity and environmental friendliness and is expected to be a new type of marine antifouling coating with desirable antifouling functions. Finally, the future research directions of antifouling coatings are prospected, which are intended to provide a reference for the development of efficient, broad-spectrum and green marine antifouling coatings.

## 1. Introduction

Fouling organisms, such as macroalgae, barnacles and shells, exist in abundance in nature. These fouling organisms can attach to and colonize a variety of surfaces, including ships, pipelines and aquaculture nets, and their attachment and accumulation on wet surfaces are known as biofouling [1,2]. Biofouling provides a range of impacts to human activities associated with the sea and has received increasing attention in recent years. As shown in Figure 1, the adverse effects of biofouling can be divided into three areas: economic, environmental–ecological and safety. On the economic side, fouling organisms attached to the surface of a ship’s hull can increase the weight and surface friction resistance of the ship, leading to higher fuel consumption. In other areas of the marine industry, fouling organisms attach to fishing nets, causing deformation of the nets and thus reducing fish production, resulting in economic losses, and, in the area of sensors used for marine monitoring, fouling organisms attach and reduce the accuracy of the sensors, in addition to increasing maintenance costs by eroding the various surfaces to which they attach, resulting in significant economic losses. The overall economic cost of marine biofouling is still reported to be over USD 150 billion per year. In environmental and ecological terms, biofouling causes ships to consume more fuel and increases greenhouse gas emissions. In terms of safety, biofouling adheres to surfaces, such as metal and concrete, and secretes biological acids that corrode materials, affecting the proper functioning of ships, marine instruments and facilities, shortening their lifespan and creating safety hazards [3,4,5].

As a result, various strategies have been proposed to prevent the adhesion of marine fouling organisms to the surface of underwater equipment and to reduce frictional resistance. The increasing trend in the publication of marine-fouling-related literature from 2009 to 2022 is shown in Figure 2, which shows that the marine fouling field has received increasing scholarly attention in the past decade. Of these, the application of antifouling coatings has become the most common means of antifouling due to their efficiency, low cost and ease of maintenance. Traditional antifouling coatings typically use biocides to kill fouling organisms, and, in the 1950s, tributyltin (TBT) was introduced as a biocide in antifouling coatings and became the dominant antifouling agent for the next half century. However, due to its toxicity to non-target marine organisms as well, TBT could also have harmful effects on other marine organisms during its service life, and, finally, TBT was banned as an antifouling agent by the International Maritime Organization in 2008 [6]. Subsequently, biocides containing heavy metal ions, such as copper and zinc, were gradually used to replace organotin antifoulants; however, studies have shown that heavy metal ions can accumulate in marine organisms and cause serious damage to marine ecosystems [7,8]. At present, people pay more and more attention to the protection of the marine ecological environment, and polluting antifouling coatings no longer meet the development needs of green oceans, so it is important to review the environmentally friendly antifouling coatings in recent years.

This paper reviews the research progress of new environmentally friendly marine antifouling coatings for ship surfaces in recent years, analyses the advantages and disadvantages of each type of antifouling coating and focuses on the preparation of antibacterial-peptide-modified surfaces. Finally, the future development direction of marine antifouling coatings is prospected, aiming to provide a reference for the development of efficient, broad-spectrum and green marine antifouling coatings.

## 2. Attachment and Fouling Mechanism of Marine Organisms

The attachment of marine fouling organisms is a highly complex process. In general, the formation of marine biofouling consists of three main stages: formation of a conditioned film, formation of biofilm and formation of the fouling biotope (Figure 3) [9]. Once the surface of any object is immersed in seawater, within a very short period, organic substances, such as polysaccharides, proteins and glycoproteins, in seawater will be deposited on the surface of the object through van der Waals forces, hydrogen bonds, electrostatic forces and other interactions to form a conditioned film (conditioning film). At this stage, the process is reversible and can provide nutrients for subsequent microbial attachment [10]. When the conditioned film is formed, microorganisms, such as bacteria and diatoms, adhere to the conditioned film within 1 day, and these attached bacteria and algae secrete to produce viscous extracellular polymeric substances (EPS) consisting of polysaccharides, proteins and glycoproteins to further enhance the attachment of microorganisms, such as bacteria and diatoms, to the surface of the material. This results in the formation of biofilm consisting of water, organic matter, microorganisms and extracellular metabolites [11]. The formation of biofilm provides abundant food and nutrients for the attachment of algal spores and some protozoa so that the surface is covered with a mucus layer within a few days. In the following weeks, a wide variety of macrofouling organisms, such as barnacles, mussels and macroalgae, will attach to the surface and grow to form complex biotopes, which will form massive fouling in a few months, resulting in a severe biofouling phenomenon [12].

## 3. New Environmentally Friendly Antifouling Coating

Since the age of navigation, people have been working to solve the problem of biofouling on the surface of ships. In the era of wooden ships, metals, such as lead and copper, were used to wrap the hull surface or apply hot pitch, tar and grease to prevent biofouling [13]. With the introduction of iron ships, the method was soon eliminated because the presence of copper accelerated the corrosion of the hull surface and the metal wrapping reduced the hull’s travel rate. By the late 18th century, other toxic substances, such as arsenic, sulfur and mercury, were introduced to protect ship hulls [14]. Since the 1950s, organotin coatings began to be widely used for marine antifouling; however, organotin was found to accumulate this chemical in many organisms, including crustaceans, fish, birds, mammals and even humans. The accumulation of organotin can cause genetic mutations in organisms and endanger human health; in addition, organotin coatings degrade slowly and can continue to accumulate in the marine environment for months to decades [15]. In 2001, the International Maritime Organization (IMO) assessed the adverse effects of organotin coatings on the marine environment and banned the use of these coatings on ship surfaces from 2008 onwards. Therefore, the development of environmentally friendly antifouling strategies has become the focus of research in the field of fouling reduction on ship surfaces, such as fouling release antifouling coatings, biomimetic antifouling coatings, photocatalytic antifouling coatings and new antifouling materials based on antimicrobial peptide preparation, etc. (Figure 4). Moreover, Table 1 provides basic information about these different antifouling strategies. These different antifouling strategies have great potential for application in the marine field.

### 3.1. Fouling Release Antifouling Coating

Compared with traditional self-polishing antifouling coatings, fouling release antifouling coatings (FRCs) do not contain antifouling agents but mainly use the dual characteristics of low-surface-energy materials with difficult surface adhesion or easy surface desorption of polluting organisms to prevent or reduce the adhesion of marine polluting organisms without releasing toxic substances into the ocean and become the focus of research on new environmentally friendly antifouling coatings because of their environmental friendliness and good antifouling properties. In 1971, Baier first investigated the relationship between surface energy and relative adhesion and proposed the famous “Baier curve” [16,17], which became the theoretical guide for subsequent low-surface-energy antifouling materials, and several studies have shown that surfaces with a surface energy of 22–25 mJ/m^2^ can exhibit the best resistance to adhesion [18]. Therefore, the widely used fouling release antifouling coatings are currently prepared mainly from low-surface-energy silicones and fluoropolymers. The most widely used organosiloxane products are polydimethylsiloxane (PDMS) elastomers, which not only have low surface energy and low modulus of elasticity but also have low surface roughness and usually have better fouling release capability compared to rigid fluoropolymers. Based on these excellent properties, PDMS has become a more mature carrier material for fouling release technology in recent years. However, unmodified PDMS has defects, such as low mechanical strength and poor bonding ability, which limit its application in the field of marine antifouling [19,20]. Current research on PDMS elastomeric antifouling coatings is mainly focused on improving the mechanical strength, increasing the bonding strength with the substrate and further improving its antifouling performance through modification [21].

Nanomaterials play an important role in the fields of environmental remediation, sustainable medicine and green energy [22,23], and studies have shown that the incorporation of nanofouling agents, such as copper oxide nanoparticles (CuO NPs), silver nanoparticles (Ag NPs) and silicon dioxide nanoparticles (SiO_2_ NPs), into a PDMS polymer matrix can effectively improve their mechanical properties and adhesion strength. Moreover, the size, morphology and chemical composition of nanomaterials can be adjusted to provide different antibacterial activity, corrosion resistance, surface hydrophobicity and mechanical properties of antifouling coatings [24]. Padmavathi et al. [25] prepared a nanocomposite with PDMS as a carrier doped with CuO, and the modified PDMS surface showed more significant inhibition of microalgae and macrofouling compared to the original PDMS surface. Liu et al. [26] developed a new silica-based nanocomposite coating consisting of a PDMS matrix and a new nanofiller of Cu@C core–shell nanoparticles. The uniformly dispersed Cu@C core–shell nanoparticles in the PDMS matrix enhanced the antifouling ability, and the introduction of Cu@C core–shell nanoparticles effectively improved Young’s modulus, tensile strength and elongation at the break of the PDMS matrix. In addition, the antifouling test results showed that the PDMS-based Cu@C core–shell nanocomposite coating exhibited excellent antifouling/decontamination performance even under static conditions. Graphene, a two-dimensional nanomaterial composed of carbon atoms, has high-performance electrical conductivity, excellent mechanical strength and thermal properties, and graphene and its derivatives, including graphene oxide (GO) and reduced graphene oxide (rGO), have been shown to have antimicrobial activity and reduce biofilm formation [27]. Selim et al. [28] used a solution-casting method to create a microstructure with a bionic PDMS substrate filled with GO-Fe_3_O_4_ coating and prepared mosquito-like PDMS/GO-Fe_3_O_4_ nanocomposites. The results showed that the coating exhibited excellent superhydrophobicity, with a water contact angle of 158° ± 2° at a GO-Fe_3_O_4_ mass fraction of 1 wt%, and exhibited excellent mechanical stability, biodegradability and antibacterial properties with the highest surface self-cleaning ability compared to pristine PDMS and other nanocomposites. Soleimani et al. [29] obtained a GOH@Ag nanocomposite by reduction of graphene oxide with *Avicennia marina*/silver (Figure 5). The results of the study showed that the PDMS coating doped with 0.5 wt% GOH@Ag had the highest RMS roughness (103 nm), the lowest barnacle-like adhesion strength (0.16 MPa) and the highest hydrostatic contact angle (118.8°), and the lowest critical surface energy (16 mN/m) after 60 days of in situ evaluation in natural seawater similarly showed that the PrG coating containing 0.5 wt% GOH@Ag showed better performance in preventing biofilm formation and fouling adhesion.

In addition, the fouling release ability of low-surface-energy materials can be effectively improved by grafting functional groups, such as amphiphiles, PEGs and quaternary ammonium salts, on PDMS or by preparing fluorosilicone co-modified materials. Wang et al. [30] designed an amphiphilic copolymer with fluorosilicone macromonomers as hydrophobic soft-side chains and amphiphiles as hydrophilic hard-side chains (Figure 6). The results showed that the coating retained plenty of low-surface-energy fluorosilicone segments on the surface even underwater, giving full play to the synergistic effect of the amphiphilic polymer while exhibiting excellent resistance to various proteins. Guazzelli et al. [31] prepared a new amphiphilic poly(oxyethylene) perfluorooctyl methacrylate (EF) consisting of PDMS methacrylate (Si) and a new amphiphilic methacrylate copolymer (Si-co-EF) antifouling coating. The long-term marine antifouling performance and adhesion strength of the Si-co-EF antifouling coatings were evaluated by marine field immersion tests, which showed that, after up to 10 months of immersion, Si-co-EF was effective in improving the fouling release performance compared to the original PDMS-coated polyvinyl chloride (PVC) panels and still maintained good adhesion on the PVC panels.

Although the low-surface-energy coating has good antifouling properties, it has poor adhesion on the hull surface, and, in practice, an intermediate coating must be used to enhance the adhesion of the low-surface-energy antifouling coating, which complicates the coating application process and increases the application cost. Moreover, compared with other types of coatings, low-surface-energy coatings are more easily damaged.

### 3.2. Natural Antifouling Agent

Traditional antifouling coatings achieve their goal of antifouling by adding heavy metal ions, but they can also have adverse effects on the ecological environment. Inspired by the antifouling ability of organisms in nature, natural antifouling agents have become a potential antifouling strategy. Natural product antifoulings are naturally active substances extracted from land plants and marine flora and fauna using biotechnology (Figure 7). They can reduce the adhesion of other microorganisms, such as bacteria, spores and larvae, by inhibiting adhesion, inhibiting metamorphosis and interfering with neurotransmission and tropism. Natural antifoulants are mostly bio-derived organic compounds, such as steroids, fatty acids, amino acids, indoles and alkaloids [32], which have better biocompatibility and degradability.

In order to better investigate the antifouling activity of natural extracts, researchers have specified the following characteristic values. LC_50_ indicates the lethal concentration, or semi-lethal concentration, which represents the concentration required to kill 50% of the experimental fouling organisms, and EC_50_ indicates the maximum effective concentration, or semi-inhibitory concentration, which represents the concentration required to inhibit the activity of 50% of the experimental fouling organisms. The ratio of LC_50_/EC_50_ is an indicator of the antifouling agent: the higher the ratio, the higher the safety. Conversely, the smaller the ratio, the lower the safety [33].

#### 3.2.1. Marine-Derived Natural Antifouling Agent

Benthic invertebrates in the ocean, such as sea cucumbers, sea squirts and sponges, can defend against external threats by secreting antifouling active substances. Darya et al. [34] tested the antifouling activity of nine bioactive extracts (non-polar to polar) from different organs of sea cucumbers against five bacterial strains, barnacles and brine shrimp larvae and demonstrated that body wall ethyl acetate extracts had the highest in vitro antifouling activity, including the results of a 3-month live sea pegboard test that showed that antifouling coatings supplemented with body wall ethyl acetate effectively inhibited the adhesion of fouling organisms, while a combination of 80 wt.% polycaprolactones (PCL) and 20 wt.% polylactic acid (PLA) with body wall ethyl acetate exhibited the highest antifouling activity. Levert et al. [35] studied five monoterpenoids (Cordiachromene A, Didehydroconicol, Epiconicol, Methoxyconidiol, Conidione) (the chemical molecular structure is shown in Figure 8) from secondary metabolites of sea squirts and the antifouling test results showed that Cordiachromene A and Epiconicol showed the strongest inhibitory effect on barnacle sedimentation and bacterial growth.

Moreover, researchers have found that some compounds isolated from marine bacteria also exhibit good antifouling activity. Pereira et al. [36] evaluated the antifouling activity of napyradiomycin derivatives isolated from marine actinomycetes, and experimental results showed that napyradiomycin derivatives can inhibit ≥80% of marine-biofilm-forming bacteria, as well as the sedimentation of *Mytilus galloprovincialis* larvae (EC_50_ < 5 µg/mL and LC_50_/EC_50_ > 15), without affecting their activity. Nong et al. [37] isolated a new cyclic tetrapeptide asperterrestide B and 11 known compounds from the marine origin fungus *Aspergillus terreus* SCSGAF0162, and the results showed that the compounds alantrypinone, methyl 3,4,5-trimethoxy-2-[2-(nicotinamido) benzamido] benzoate and penicillixanthone exhibited strong antifouling activity against amphibious small-vine larvae (EC_50_ values of 17.1 ± 1.2, 11.6 ± 0.6 and 17.1 ± 0.8 µg/mL, respectively).

#### 3.2.2. Terrestrial-Derived Natural Antifouling Agent

In addition, terrestrial plants are also the focus of scholars’ research. Feng et al. [38] investigated the antifouling properties of 18 alkaloids extracted from terrestrial plants, and a 1-year marine field antifouling study showed that camptothecin could significantly inhibit the adhesion of marine fouling organisms. Subsequently, the degradability and environmental risks associated with the use of camptothecin as a novel marine natural antifouling agent under marine environmental conditions were further evaluated [39], which showed that camptothecin can be rapidly decomposed under light, while the mean environmental concentration (PEC)/no effect concentration (PNEC) values of camptothecin were well below 1, indicating that the environmental risks associated with the use of camptothecin as a marine antifouling agent were low. This naturally active substance has excellent application prospects in the field of marine antifouling. Selvaraj et al. [40] identified a new carotenoid, lycopene, from mangrove extracts and found it to have excellent inhibitory ability against marine macrofauna fouling proteins.

In complex marine environments, the effectiveness of a single antipollution strategy may be limited. Based on collaborative strategies, combining natural antifouling active substances with other antifouling functional materials can effectively improve their applicability in marine environments. Chen et al. [41] prepared a new green low-surface-energy antifouling coating UBCP/PTO by compounding natural-product-polymerized tung oil (PTO) with urushiol-based benzoxazine copper polymer (UBCP) (Figure 9A). The composite coating has low elastic modulus and low surface free energy, which can effectively inhibit the adhesion of fouling microorganisms on the surface, and a small number of copper ions in UBCP can be released from the composite coating with a synergistic antifouling effect. Zmozinski et al. [42] investigated the antifouling properties of silicone resins supplemented with rue and ginger oleoresin extracts. Bacterial adhesion tests showed that the rue extract was effective in reducing the adhesion of *Escherichia coli* compared to ginger oleoresin, and, after 6 months of marine field immersion tests, the coatings containing rue and ginger oleoresin extracts were effective in inhibiting the adhesion of marine fouling organisms (Figure 9B).

#### 3.2.3. Synthetic Natural Antifouling Agent

However, there are still some challenges in the application of natural antifouling agents, such as the low content in organisms, the insufficient separation and purification technology, which makes it difficult to obtain large quantities, as well as the poor stability and the difficulty of long-term preservation. In recent years, researchers have undertaken a series of attempts to further improve the applicability and broad spectrum of natural antifouling agents, such as synthesizing similar antifouling active substances artificially by chemically modifying natural antifouling active substances, introducing bactericidal functional groups or compounding with antifouling agents. Labriere et al. [43] synthesized a natural antifouling active substance analog by chemically modifying the marine natural product Phidianidine A (1) (the chemical molecular structure is shown in Figure 8) and conducted 84 days of antifouling tests on real sea pegboards. The results showed that the antifouling coating added with Phidianidine A (1) analog had a strong inhibitory effect on the settlement of marine bacteria and microalgae. Indole compounds derived from marine organisms have been shown to have excellent antifouling activity and excellent biocompatibility, and the application of this natural antifouling agent in antifouling coatings has been much reported in recent years [44,45]. Ni et al. [46] synthesized an indole derivative by Friedel-Crafts alkylation reaction (Figure 10); the results of the study showed that the synthesized indole derivative could effectively inhibit the attachment of different bacteria and algae. In addition, the results of the 6-month marine experimental pegboard likewise showed that the antifouling properties of the acrylic metal salt resin containing the indole derivative were superior to those of the pure resin, and the introduction of the indole derivative further improved the antifouling properties of the multifunctional acrylic metal salt resin.

The secondary metabolite xanthone derivatives, which are widely present in marine and terrestrial plants and animals, have been shown to have antifouling activity yet are still not used for marine antifouling. Resende et al. [47] explored the antifouling activity of a series of 24 synthetic xanthone analogs for the first time to further explore the potential of xanthone derivatives for antifouling. The test results showed that xanthones 21 and 23 (the chemical molecular structure is shown in Figure 8) were the most effective in inhibiting larval sedimentation (EC_50_ of 7.28 and 3.57 µM, respectively), and, with a therapeutic ratio (LC_50_/EC_50_) > 15 for xanthone 23, this synthetic analog has great potential for marine antifouling applications.

### 3.3. Micro/Nanostructured Antifouling Materials

Some plants and animals, such as sharks, sea crabs, lotus leaves and rice leaves, exhibit excellent self-cleaning and antifouling and deterrent abilities on their surfaces, which can effectively prevent or inhibit the adhesion of fouling organisms, such as barnacles, algae and bacteria [48]. Recent studies have revealed that various unique micro/nanostructures on the surfaces of these plants and animals play an important role in antifouling and deterrence, such as the shield-shaped scales in striped grooves on the surface of shark skin, the dense microprotrusions on the surface of lotus leaves and the rounded raised structures on the surface of sea crabs. Scardino [49,50] proposed the famous “attachment point theory” based on extensive experimental studies; that is, fouling organism larvae tend to attach to surfaces close to their body surface size, and this theory became an important theory for subsequent research on micro/nanostructured antifouling materials. This theory has become an important theoretical guide for subsequent research on micro/nanostructured antifouling materials. Many methods have been applied to fabricate micro/nanostructured surfaces over the years, such as deposition methods, templating or soft lithography, etching processes and nanocomposite methods [51], and these methods have aided the application of micro/nanostructured materials in marine-antifouling-related fields.

Sharks are one of the fastest swimming organisms in the ocean, and studies on their skin have revealed that the surface of shark skin is arranged with a finely ribbed groove structure in the direction of the current, which enables sharks to exhibit significant drag reduction in the current, and the non-smooth groove structure surface of shark skin also has better antifouling bioadhesive properties, which breaks the traditional notion that, the smoother the surface, the lower the drag, and reveals a new notion of microstructure antifouling [52]. Carman et al. [53] designed and prepared a sharkskin-like microstructured surface Sharklet AF^TM^ on PDMS substrate, which could effectively inhibit the settlement of 86% of marine algal spores when the size of Sharklet AF^TM^ was smaller than the size of marine algal spores. Qin et al. used the idea of segmental independence to simplify the structure of biological sharkskin shield scales [54] and then built a non-smooth bionic structure on the PDMS surface by die-casting and further modified the bionic non-smooth surface using zeolite-type imidazole acid skeleton-67 (ZIF-67) particles, and the adhesion rate of *Chlorella vulgaris* on the bionic non-smooth surface decreased by 65.3% compared to the smooth surface. He et al. [55] prepared poly (ionic liquid) brush-grafted “Sharklet” surfaces by 3D printing technique and subsurface-initiated ROMP (Figure 11A). The experimental results showed that p[BNIm][Br] exhibited good anti-bovine serum protein adsorption and better antibacterial properties against *E. coli* and *S. aureus*. The surface and chemical composition of “Sharklet” could inhibit the attachment of microalgae through synergistic effects. Further, p[BNIm][Br] grafted on the surface of “Sharklet” increased the percentage of removal of *Porphyridium* from 40.7% to 60.1%. The aforementioned study successfully inhibited the deposition of fouling organisms by constructing sharkskin-like microstructures with constant height on the material substrate; however, the grooved structures on the sharkskin surface did not have a constant height and therefore lacked a certain degree of biological accuracy. Munther et al. [56] further improved on the existing sharkskin-like microstructures by generating microstructures on PDMS substrates with integrated height gradients, increasing the complexity of the surface microstructure, and using more mimic microstructure surfaces with integrated height gradients can further reduce the attachment of fouling organisms compared to microstructure features with constant height.

Unlike sharks that swim at high speed in seawater while secreting mucus, researchers have found that some relatively stationary animals and plants in the ocean also exhibit inhibition of fouling organisms on their surfaces during high and low fouling seasons, such as mussels and kelp in the ocean. Guan et al. [57] used a mussel-like surface “blossom tree” micro/nanostructure inspired by mussels and evaluated the anti-diatom performance of this mimic micro/nanostructure using three diatom species, and the results showed that the “flowering tree” micro/nanostructure could effectively inhibit the attachment of diatoms. Zhao et al. [58] constructed the microstructure morphology of a *Laminaria-japonica*-like surface on PDMS substrate; at the same time, sodium alginate and (guanidine-hexamethylenediamine-PEI) (poly (GHPEI) were alternately deposited on the surface of biomimetic materials by layer by layer assembly method, forming a thin and highly hydrated polymer film, further modifying the structured surface (Figure 11B). The antifouling test results show that the modified biomimetic surface can effectively inhibit the adhesion of *Nitzschia closterium* and *Escherichia coli*.

The morphology of micro/nanostructures on the surface of terrestrial animals and plants has also been studied by scholars. Shahali et al. [59] conducted a systematic study of three different Australian cicada wing structures, *Psaltoda Claripennis* (PC), *Aleeta curviest* (AC) and *Palapsalta eyrie* (PE), and confirmed that the three cicada structures killed bacteria by disrupting the cell membranes of attached bacteria, while human osteoblasts attached to the cicada surface remained intact cellular morphology after 24 h, indicating that the cicada has excellent biocompatibility. Subsequently, titanium nanopillar arrays with cicada-wing-like structures were successfully prepared on silicon wafers using electron beam lithography, which were observed to disrupt the bacterial cell membrane of *P. aeruginosa* in a cicada-wing-like manner and maintain good compatibility with human osteoblasts. Du et al. [60] constructed six types of bionanostructures (sharkskin-like, rose-leaf-like, piggyback-like, rice-leaf-like, butterfly-wing-like and bamboo--leaf-like) on zirconium-based bulk metallic glasses (Zr-BMGs) using the technical means of femtosecond laser (Figure 11C). The antifouling performance of different types of bionanostructures on the sample surface under different laser intensities was investigated. The results showed that the bionanostructured surface had better antifouling properties than the polished surface, and bacteria were able to make complete contact with the polished surface, while the sample surface constructed with bionanostructured microstructures changed the contact state between bacteria and the material, thus achieving the effect of inhibiting bacterial adhesion. Among the six biomimetic structures, the rice-leaf-like structure and the butterfly-wing-like structure had the best antifouling properties to inhibit bacterial adhesion, while the sharkskin-like structure had the best bactericidal properties. In addition, the butterfly-wing-like structure processed with 0.23 J/mm^2^ laser energy intensity had the best balance between antifouling, bactericidal and cytotoxicity.

In addition to the above-mentioned surface microstructures imitating natural organisms, domestic and foreign researchers have prepared various regularized artificial surface microstructures to investigate their antifouling mechanisms and properties. The authors’ group designed and processed six microstructured surfaces in a previous study [61] and then prepared a synergistic surface by peptide modification. The experimental results showed that six synergistic surfaces with a depth of 0.8 μm were successfully prepared on the 304 stainless steel substrate surface, and the contact angle of the sample surface increased from 75° to 116.99°. In addition, the antifouling test results showed that the synergistic surface had strong anti-algal properties, and the anti-*Chlorella pyrenoidosa*) and anti-*Phaeodactylum tricornutum* rates reached 78.56% and 87.80%, respectively, after 7 days of immersion in artificial seawater.

Compared with the traditional antifouling technology of releasing antifouling agents to kill marine fouling organisms, micro/nanostructured antifouling materials mainly use their microstructure morphology to inhibit the adhesion of fouling organisms and do not cause damage to the marine ecological environment. However, most of the micro/nanostructures are processed on soft materials, such as PDMS, silicon wafers, etc. Only a small amount of research has focused on microstructure antifouling of hard metal materials used in ships, and there are also disadvantages, such as complicated processes and high production costs of imitating biomimetic micro/nanostructures. All the above factors restrict the practical application of micro/nanostructured materials in the field of marine antifouling.

### 3.4. Hydrogel Antifouling Material

In recent years, while studying the surface micro/nanostructure of plants and animals, scholars found that the mucus secreted by the epidermis of some fish and amphibians can effectively resist the adhesion of fouling organisms. Further studies have revealed that the main component of the mucus is a natural hydrogel for mucin, which is similar to synthetic hydrogels in that mucin has soft and hydrophilic properties and can form gel properties in water. Hydrogels contain cross-linked three-dimensional polymer networks that can absorb large amounts of water and are hydrophilic, which can provide favorable conditions for subsequent hydrogen bonding or electrostatically induced hydration layer formation, which can effectively inhibit the adhesion of fouling organisms in the ocean [62]. Because of these advantages, researchers have conducted extensive research on the application of hydrogel materials in marine antifouling [63,64,65]. However, in marine antifouling applications, it is still a challenge to apply hydrogel coatings to various surfaces with high density stably and conveniently, and, in addition, weak adhesion strength and poor durability greatly limit the practical application of hydrogel coatings.

In response to these problems, scholars have carried out a great deal of work on modified hydrogel coatings. As discussed in the previous subsection, nanomaterials have excellent properties; the same nanocomposite polymer hydrogels prepared based on doped nanoparticles can effectively improve their mechanical strength. Moreover, the method is easy to operate, inexpensive and convenient for mass production. For example, Zhang et al. [66] prepared a hydrogel-anchored iron-based amorphous (HAM) coating with excellent antifouling and corrosion protection properties (Figure 12A). The mechanical strength and corrosion resistance of the hydrogel were effectively improved by the addition of SiO_2_ nanoparticles, and the interfacial engineering strategy of micrographic patterning, surface hydroxylation and dopamine interlayer was used to significantly improve the bonding rate between the hydrogel and the amorphous coating. In laboratory and marine field environmental tests, the HAM coatings exhibited excellent antifouling properties with 99.8% resistance to algae and 100% resistance to mussels.

In addition, researchers have used chemical cross-linking methods to prepare hydrogels to improve their mechanical stability and antifouling activity, such as polyethylene glycol (PEG), polyacrylamide (PAM) and acrylate hydrogels, which can be used to obtain hydrogel coatings with suitable properties by adjusting the preparation parameters. Yang et al. [67] successfully prepared a coatable anti-sewage gel coating based on polyacrylamide-cross-linked dobby polyethylene glycol (Figure 12B), which can cover various substrate surfaces uniformly and densely, and the epoxy intercoat introduced can provide strong non-covalent adhesion to various surfaces and covalent bonding to the hydrogel layer, thus greatly improving the adhesion ability and mechanical stability of the hydrogel coating, and the hydrogel coating also exhibited excellent antifouling properties. Lu et al. [68] prepared a bio-based amphiphilic hydrogel coating with excellent antifouling and mechanical properties. The coating was achieved by in situ formation of hydrophobic and hydrophilic interpenetrating polymer networks (IPN). The hydrophobic part of the synthesized silicone containing epoxy resin had excellent mechanical properties, including high tensile strength and excellent adhesion, while the hydrophilic hydrogel part of the hydrophilic polymer cross-linked with AgNPs had excellent antifouling properties. Further, 45 days of real sea field tests showed that the bio-based amphiphilic hydrogel coating could effectively inhibit marine fouling. The amphiphilic hydrogel coating (SA-1-5) loaded with sulfhydryl hydrophilic polymer (PNIPAM-SH) and a silver salt of trifluoromethane sulfonate (AgOTf) showed the best fouling resistance. This is because, when the amphiphilic hydrogel coating is immersed in water, the hydrophilic PNIPAM-SH can transfer to the surface of the coating and form a hydration layer to prevent the adhesion of fouling organisms, such as proteins, bacteria, and algae. Meanwhile, with the migration of PNIPAM-SH, AgNPs are also transferred to the surface, effectively killing marine organisms and further improving the antifouling performance (Figure 12C).

In addition, hydrogels can be used not only as antifouling coatings but also as bioactive polymer matrices to capture live bacteria, and selected bacteria produce signaling molecules that can control the subsequent adhesion and adherence processes of contaminating organisms [69], a property that broadens the application of hydrogel coatings in the field of marine antifouling. However, the application of hydrogels is still subject to extensive research testing. The antifouling test cycle for currently developed hydrogels is typically 1 to 3 months. Commercial antifouling coatings typically retain their antifouling ability for more than a few years. Therefore, longer offshore antifouling tests should be conducted.

### 3.5. Photocatalytic Antifouling Materials

Photocatalysis is an advanced oxidation technology that decomposes seawater and dissolves oxygen through the redox ability of semiconductor photocatalysts under light conditions, resulting in the generation of large amounts of reactive oxygen species (ROS), such as superoxide radicals (•O_2_^−^), singlet oxygen (^1^O_2_) and hydroxyl radicals (-OH). These ROS can react strongly with components such as lipids, polysaccharides and proteins in bacterial cell membranes, causing cell membrane rupture and thus leading to bacterial death [70]. Therefore, photocatalytic antifouling materials have good characteristics, such as low toxicity and environmental friendliness, and have good prospects for application in the field of marine antifouling. Previous photocatalytic coatings are mainly based on titanium oxide (TiO_2_) and zinc oxide (ZnO), which show good chemical stability and antifouling properties under UV conditions. However, the visible light utilization of these coatings is very low [71]. At present, the research on photocatalytic coatings is mainly focused on improving photocatalytic activity and visible light utilization.

CeO_2_ has excellent properties, such as abundant reserves, low price, high stability and non-toxic to mammals, and has been widely used as oxygen storage materials, UV absorbers and metal preservatives in recent years; however, the weak visible light absorption, wide band gap (2.8–3.2 eV) and rapid complexation of photogenerated electrons and holes of CeO_2_ have limited the application of CeO_2_ in the field of antibacterial and antifouling [72]. BiOI is a novel semiconductor material in ternary oxyhalide BiOX (X = Cl, Br and I) with excellent optical properties, high chemical stability, non-toxicity, corrosion resistance and other favorable characteristics. Moreover, BiOI can be used to modify wide bandgap semiconductor materials to improve photocatalytic efficiency by forming heterojunctions [73,74]. Mao et al. [75] prepared a flower-like BiOI@CeO_2_@Ti_3_C_2_ ternary heterojunction by hydrothermal method, which can effectively improve the photocatalytic properties of CeO_2_ and facilitate the application of CeO_2_ in the field of marine antifouling. When the mass ratio of CeO_2_@Ti_3_C_2_ to BiOI is 10, the prepared ternary heterojunction (BCT-10) exhibits the best photoelectrochemical and antibacterial properties, and the photocatalytic bacterial inhibition efficiency for *E. coli* and *S. aureus* can reach 99.76% and 99.89%, respectively, which are 2.98 and 3.07 times higher than that of pure CeO_2_. Under the joint action of the two heterojunctions, BCT-10 can effectively promote photogenerated electron–hole transfer and increase the generation of ROS through oxidative stress (Figure 13A), thus enhancing the photocatalytic antibacterial effect of CeO_2_. The successful preparation of ternary heterojunctions provides a new research idea for the application of CeO_2_ in marine antibacterial and antifouling fields.

Cu_2_O has limited its application in marine antifouling because of its drawbacks, such as toxic risk to environmental pollution and inability to generate ^1^O_2_ under light, and, in recent years, studies have shown that the p–n junction heterostructure of inorganic–organic semiconductors can achieve precise concentration control, full solar spectrum driving and ^1^O_2_ generation [76]. Polyethylene-3,4,9,10-tetramethylbenzidine (PDINH) is widely used because of its excellent photothermal stability, full solar spectrum absorption and carrier mobility [77]. Therefore, by rational design of Cu_2_O/PDINH heterostructures, the toxicity risk of Cu_2_O to environmental pollution can be reduced, full solar-spectrum-driven sterilization can be achieved and the inherent defect that Cu_2_O cannot produce ^1^O_2_ can be overcome. Ma et al. [78] constructed an inorganic–organic photocatalytic antimicrobial agent Cu_2_O/PDINH by integrating p-type Cu_2_O and n-type PDINH heterostructure, which successfully increased the photoavailability by 75% and exhibited excellent long-term and photocatalytic antimicrobial activity, with the inhibition rates of *S. aureus* and *P. aeruginosa* remaining above 90% after 30 days, and the results of antifouling tests also showed that the Cu_2_O/PDINH antifouling coating had almost no dirt adherence after 60 days of immersion. The antibacterial mechanism of the Cu_2_O/PDINH heterogeneous structure is the synergistic effect of the toxicity of copper ions and the continuous production of ROS driven by the full solar spectrum (Figure 13B), so this inorganic–organic Cu_2_O/PDINH heterogeneous structure has great potential for environmentally friendly marine antifouling agent applications.

Although photocatalytic antifouling materials have good characteristics, such as low toxicity and environmental friendliness, the above scholars have used different methods to improve the application of photocatalytic materials in the field of marine antifouling. However, a great deal of research needs to be conducted to improve the photocatalytic activity of photocatalytic materials from several aspects, such as the compounding of electron–hole pairs in photocatalytic materials, expanding the light utilization rate of photocatalytic materials and increasing the reaction between photocatalytic materials and reactants [79]. In addition, photocatalytic materials are more dependent on light, and the performance of photocatalytic antifouling materials will be significantly reduced under low light or dark conditions, so it is a further research direction to study alternative materials that can maintain antifouling performance under low light and dark conditions.

### 3.6. Slippery Liquid-Infused Porous Surfaces

Inspired by the special microstructure and smooth properties of natural pigweed, researchers have prepared slippery liquid-infused porous surfaces (SLIPS) with a stable continuous lubrication layer by injecting low-surface-energy lubricants into substrates with micro/nanoporous structures. The surface has excellent hydrophobicity, low friction and non-adhesive and self-healing properties. The broad application prospects of SLIPS in the fields of corrosion protection, anti-icing, antibacterial and antifouling make it a hot topic in current bionanomaterials research.

However, SLIPS are susceptible to environmental shear forces, such as ocean flow or foreign fluids, leading to the destruction of porous structures and loss of surface lubricants, thus depriving SLIPS of their ability to protect substrates from contamination. Therefore, finding ways to improve the stability and extend the service life of SLIPS is important for their application in marine antifouling. Tong et al. [80] prepared a smart SLIPS marine antifouling coating with responsive switching lubrication mode and self-healing properties inspired by the defensive behavior of blind eels that secrete mucus to escape from predators. The responsive supramolecular interaction between azobenzene (Azo) and α-cyclodextrin (α-CD) can regulate the lubricity of SLIPS; thus, the SLIPS marine antifouling coating can flexibly switch the antifouling mode under different environmental conditions, which greatly extends the service life in the marine environment, and the introduction of disulfide and hydrogen bonds effectively enhances its self-healing property (up to 91.73%). Moreover, this antifouling coating has the longest antifouling time compared to the reported SLIPS materials (at least 180 days of antifouling performance in real marine field tests). Li et al. [81] prepared SLIPS with dual protection (anti-corrosion and antifouling) on aluminum substrates by fabricating regular microstructure arrays with hydrophobic properties on PDMS coatings based on photolithography and then injecting perfluoropolyether lubricants to improve the durability and stability of SLIPS. The microstructure array can store more lubricant and improve the stability of the lubricant layer; in addition, even if the lubricant layer fails, the PDMS coating (second layer) can further block the infiltration of corrosive liquid, and the low surface energy of the PDMS coating can help to inhibit the adhesion of fouling organisms; the antifouling test results also show that the SLIPS can effectively inhibit the adhesion of algae.

The quaternary ammonium salt surface can bind to negatively charged bacterial cell membranes through polar (coulombic) interactions, while the hydrophobic tail of the surfactant penetrates the hydrophobic interior of the cell membrane and disrupts the bacterial integrity [82]. Therefore, the QAS ionic liquid-injected surface for in situ growth of MOF is expected to address the need for stable storage of sufficient lubricant and the effectiveness of long-term antifouling due to insufficient thickness. Li et al. [83] used trimeric acid (H_3_BTC) solution and Al flakes to synthesize MIL-110 porous surface in situ in one step hydrothermally (Figure 14). The results of both 10- and 21-day antifouling tests showed that the SLIPS had excellent antifouling properties. The adsorption of lipopolysaccharide (LPS) by SLIPS was 50% lower than the adsorption of LPS by aluminum plates and aluminum plates with MIL-110 grown on the surface within 3 h.

Although SLIPS materials have proven to be a promising application solution for the biofouling of marine equipment surfaces, concerns have arisen in recent years about the impact of lubricant release from SLIPS materials on the marine ecosystem. Unlike banned TBT, which is extremely toxic to non-target organisms, lubricants such as perfluoro polyethers and silicone oils may trap and smother organisms on the coating, resulting in physical–mechanical effects [84], so the development of suitable lubricant alternatives is a further research direction for SLIPS in the future.

## 4. Mechanism of Action of Antimicrobial Peptides and Preparation of Modified Surfaces

### 4.1. Source and Mechanism of Action of Antimicrobial Peptides

Antimicrobial peptides (AMPs) are an important component of the natural immune system of living organisms and are widely found in nature, with a molecular composition usually less than 100 amino acids, and have a variety of biological activities, such as antibacterial and antifungal [85]. In 1972, the Swedish scholar Boman et al. [86] first identified and reported a class of peptide molecules with antibacterial activity through studies on Drosophila. Subsequently, researchers obtained the first true AMPs, cecropin, from *E. coli*-stimulated *S. cherubis*. It has been recorded that more than 2000 antimicrobial peptides have been reported in the literature.

Since the discovery of antimicrobial peptides, much research has been conducted on the antimicrobial mechanism of antimicrobial peptides. It is now known that antimicrobial peptides kill bacteria by interacting with phospholipid components to disrupt the integrity of bacterial cell membranes, and different classes of antimicrobial peptides have different mechanisms of action on cell membranes, and once the antimicrobial peptides bind to the cell membrane, there are three main models of cell membrane disruption: barrel, ring pore or carpet (Figure 15) [87,88]. In the “barrel” model, when the antimicrobial peptide reaches a critical threshold concentration, AMPs (usually α-helical peptides) are inserted vertically into the phospholipid bilayer in the manner of barrel plates, causing cellular contents to flow out and leading to cell death (Figure 15I), such as the antimicrobial peptides Magainin 2 and MSI-78. The “ring pore” model is similar to the “barrel” model. When the ratio of AMPs to lipid molecules reaches a certain value, AMPs are inserted vertically into the phospholipid bilayer and remain bound to the phospholipid head, forming a circular pore-shaped transmembrane channel that disrupts the cell’s transmembrane potential and osmoregulatory function, inhibiting cellular respiration and eventually leading to the death of the bacterium (Figure 15II). The difference between the two models is that, when the antimicrobial peptide is embedded vertically in the lipid membrane, in the “barrel” model, the hydrophilic part of AMPs faces the cavity in the middle of the “barrel” cluster, independent of the polarity of the lipid membrane. In the “ring pore” model, the hydrophilic part of AMPs always interacts with the polar head on the lipid membrane. In the “carpet” model, AMPs are folded to form an amphiphilic conformation and arranged parallel to the membrane surface as a carpet. When the antimicrobial peptide reaches a certain concentration, AMPs form micelles and partially split the lipid membrane, thus disrupting the phospholipid bilayer and promoting cell membrane disintegration and extracellular extravasation (Figure 15III), such as some linear α-helical AMPs (Magainins, Cecropin and Dermastin), the latter phase of which is sometimes referred to as the “detergent” model (Figure 15IV). In addition, there are less common mechanisms of action associated with cell membrane disruption. For some antimicrobial peptides, it has been suggested that, when a critical threshold concentration is reached, these peptides are inserted into the cell membrane as aggregates (Figure 15V). It has also been shown that some antimicrobial peptides do not penetrate the membrane but only cross the major intracellular targets they reach, such as DNA-binding Buforin II. However, at high concentrations, most membrane-bound peptides are expected to cause membrane leakage, including Buforin II. In addition, investigators have described an “electroporation” model, in which the charge of the peptide accumulated on the outer surface of the cell membrane creates a high enough potential difference to cause pores to form in the cell membrane (Figure 15VI), unlike other models in which these pores are not aligned with the antimicrobial peptide. Some antimicrobial peptides have also been shown to alter the distribution of phospholipids in the cell membrane (Figure 15VII), and the aggregation of certain phospholipids can alter the local curvature of the cell membrane, leading to phase separation, making these phospholipids unavailable for other interactions, or altering the thickness of the membrane.

### 4.2. Preparation of Antimicrobial-Peptide-Modified Surfaces

Based on the excellent antimicrobial properties and biocompatibility of antimicrobial peptides, the use of antimicrobial peptides to modify the surface of materials can generate a new material that is non-toxic, non-polluting and green. Scholars at home and abroad have developed many methods to bind antimicrobial peptides on material surfaces, which can be generally classified as the polymer brushing method, layer stacking method (LBL), monomer self-assembly method (SAM) and chemical coupling method [89,90]. The use of these methods can provide theoretical guidance for the binding of antimicrobial peptides on the surface of materials.

Polymer brush method, also known as the polymeric molecular brush method, is a modification means to covalently immobilize antimicrobial peptides onto the surface of materials through reactive groups at the end of polymeric molecules after activation [91], and some functionalized polymer resins, such as polyethylene glycol (PEG) or other ‘brushes’ with reactive groups suitable for covalent immobilization of peptides, are commonly used. This approach allows the rapid and free orientation of bound AMPs at the implant–medium interface, which can facilitate peptide–bacterial interactions and enhance the protective properties surface of the bound peptide layer on the implant. Li et al. [92] based a poly[*N*,*N* dimethyl acrylamide-co-*N*-(3-aminopropyl)methacrylamide hydrochloride] (PDMA-co-APMA) polymer brush to link antimicrobial peptides Cys-HHC-36 linked to hydroxyapatite (HA) nanorods (Figure 16A), and the bound antimicrobial peptide retained sterilizing activity against *S. aureus* and *E. coli*, being able to kill 99.5% of *S. aureus* and 99.9% of *E. coli*, which was attributed to the synergistic effect of the destruction of AMP derivatives and the physical penetration of HA nanorods. However, the polymers used to link the antimicrobial peptides are prone to degradation reactions leading to chain breakage and thus the release of the bound AMPs, and recent studies have found that polymer degradation can be minimized by using stabilized polymers (i.e., polymer-containing additives that prevent polymer degradation reactions) [91], which is important for maintaining the long-term antimicrobial properties of the material surface.

Layer stacking (LBL) is a versatile approach for the preparation of coatings on substrate surfaces, where the binding of antimicrobial peptides to the surface is achieved through electrostatic, hydrogen bonding and covalent interactions between polycations and polyanions. Shukla et al. [93] successfully doped and released AMP (xanthophyll G1) in films assembled based on the LBL method (Figure 16B), demonstrating the sustained release in LBL films and control of AMP loading. The polyanionic films were able to strongly influence the growth and degradation properties of the films as well as the incorporation and release properties of xanthophyll G1. The results of antimicrobial experiments showed that surfaces deposited with such films could effectively inhibit the growth of *S. aureus*. Plácido et al. [94] prepared a polyanionic film using the negative polyelectrolyte cashew gum (CG) and then further deposited the positive polyelectrolyte PcL342-354C polypeptide (polycation) on the glass surface based on the electrostatic interaction between the polyelectrolytes and LBL thin film. The experimental results showed that the LBL films prepared on the glass surface exhibited excellent antibacterial activity against *E. coli*.

Monomer self-assembly is another important modification method to covalently immobilize antimicrobial peptides onto the material surface. Ye et al. [95] self-assembled single-molecule membranes (SAM) based on thiol functionalization (alkyl, carboxylic acid, amine) to modulate the surface charge and/or surface polarity of gold sensor surfaces and studied the amphiphilic peptide GL13K with the substrate surface in real-time by quartz crystal dissipation monitoring microbalance (QCM-D). The results suggest that the main factor affecting AMP/substrate hydrogen bonding interactions is the surface polarity of the SAM-coated sensors rather than their surface charge, and the findings from this study can aid in the design and optimization of AMP coatings on metal surfaces. Parreira et al. [96] used a monomer self-assembly method to develop a monolithic biomimetic biotin poly (ethylene glycol) 11-maleimide spacer via a heterobifunctional (EG11-MAL) to immobilize the antimicrobial peptide MSI-78A on gold surfaces (Figure 16C); the reaction was achieved through the combination of the maleimide group (MAL) at the end of EG11-MAL and the -SH group of AMPs. The results of antimicrobial experiments showed that the surfaces obtained by this method were highly effective in inhibiting *H. pylori*.

The chemical coupling method is a method of grafting antimicrobial peptides onto the surface of a material using a chemical coupling agent. Dopamine is highly susceptible to oxidation in weak alkaline solutions, forming a composite film of polydopamine (PDA) with strong adhesion, and, in addition, many functionalized functional groups, such as catechols, amines and imines, are present on the surface of the PDA-modified materials, which can be used as reactive groups for further functionalization and the design and acquisition of the desired functional materials [97]. Therefore, dopamine has been used as a common coupling agent for surface modification of materials in recent years. Lim et al. [98] used dopamine as a coupling agent to graft biopeptides onto the sample surfaces and obtained effective antimicrobial surfaces. In a previous study by the authors’ group, which discussed in detail the preparation of antimicrobial-peptide-modified metal surfaces and their antifouling properties, the use of dopamine as a coupling agent was effective in grafting antimicrobial peptides onto stainless steel surfaces, resulting in modified sample surfaces with excellent antifouling properties [99,100,101].

In summary, to apply antimicrobial peptides to the field of marine antifouling, stable and efficient grafting methods on the surface of the substrate must be realized. Although some antimicrobial peptides in nature can spontaneously bind to the surface of the substrate, few natural peptides have both antimicrobial functional fragments and substrate affinity fragments. The antimicrobial peptides are indirectly immobilized onto the substrate surface by the above-mentioned methods, but different immobilization strategies have a great impact on the conformation and activity of the antimicrobial peptides. It remains a challenge to ensure that the antimicrobial peptide immobilized onto the surface still reflects the same spatial structure and antimicrobial properties as in the solution. In addition, there is a wide variety of antimicrobial peptides, and current research is not satisfied with peptides extracted from organisms but hopes to provide different physical properties and antimicrobial activity to biological peptides through synthetic designs, which can modulate the physical properties of metal surfaces after reacting with metals to obtain better antifouling effects.

## 5. Conclusions and Prospects

As people pay more and more attention to the ecological environment, it is important to develop antifouling coatings with stable, durable and environmentally friendly characteristics. Based on the mechanism of marine biofouling, this review discusses new environmentally friendly coatings for ship surface antifouling, focusing on the mechanism of action of antimicrobial peptides and the preparation means of modified surfaces, and finally indicates the following prospects for the follow-up research of marine antifouling coatings.

(1)The currently developed antifouling coatings have difficulty covering the problems of stability, toxicity and cost of use, which seriously hinders the promotion of their use in the marine field. In the future, the research and application of marine antifouling coatings will develop in the directions of efficient, broad-spectrum, non-toxic, non-polluting and degradable.(2)Several antifouling coatings reviewed in this paper are derived from natural inspiration, such as natural antifouling agents, micro/nanostructured antifouling materials, liquid-infused smooth porous antifouling materials and hydrogel antifouling coatings, which have the advantages of antifouling performance while remaining non-toxic and ecologically harmless, etc. However, these antifouling coatings still generally have the disadvantages of high costs and poor stability. Therefore, it is necessary to conduct in-depth research on the antifouling mechanism of plants and animals to develop efficient, broad-spectrum, non-polluting antifouling coatings.(3)The marine environment is complex, and the living environment of each fouling organism is different, so it is not possible to achieve the expected antifouling effect by only relying on coatings with a single antifouling mechanism. Therefore, the future trend is to develop antifouling coatings with synergistic strategies.

## Figures and Tables

**Figure 1 biomimetics-08-00200-f001:**
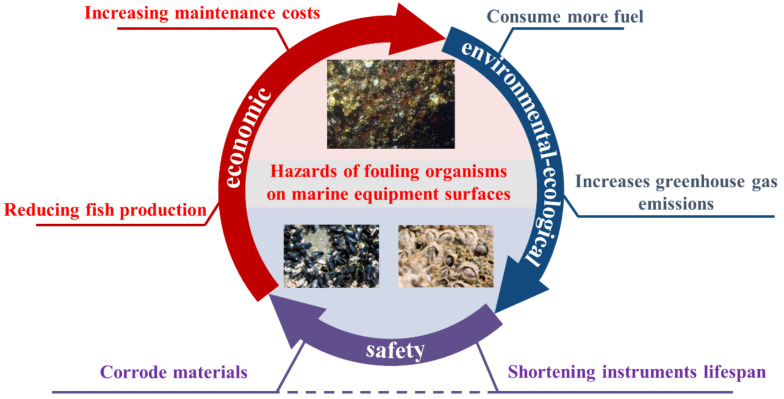
Hazards caused by fouling organisms adhering to the surface of marine equipment.

**Figure 2 biomimetics-08-00200-f002:**
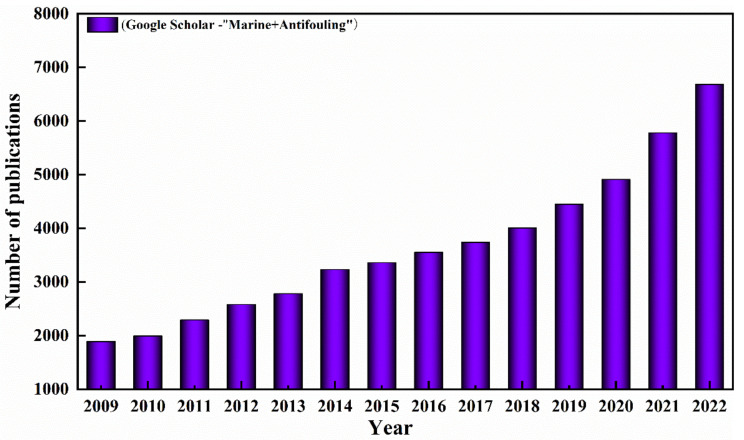
The number of publications on marine antifouling research per year from 2009 to 2022, collected by Google Scholar using the keywords “Marine + Antifouling”.

**Figure 3 biomimetics-08-00200-f003:**
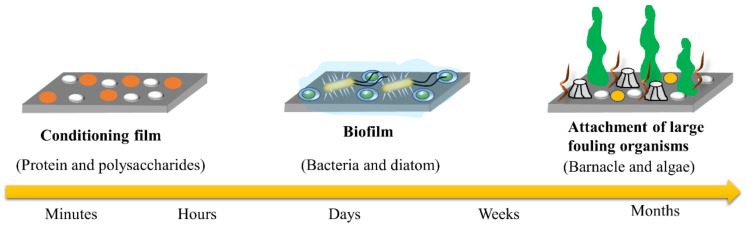
Schematic diagram of marine biofouling process.

**Figure 4 biomimetics-08-00200-f004:**
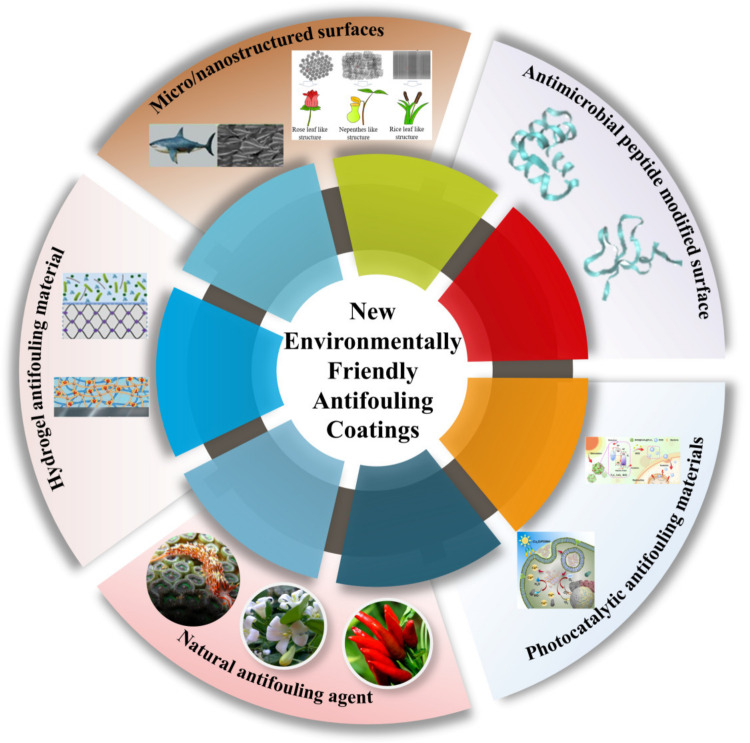
A review of new environmentally friendly antifouling coatings.

**Figure 5 biomimetics-08-00200-f005:**
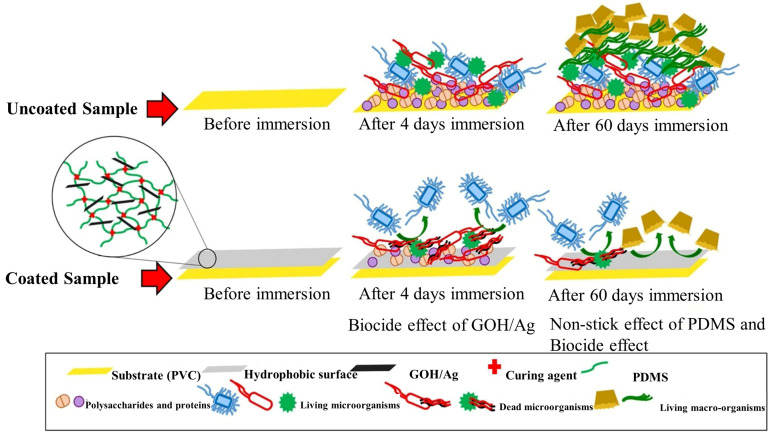
The structure of GOH@Ag nanocomposite and antifouling performance [29] with permission (Copyright © 2023, Elsevier).

**Figure 6 biomimetics-08-00200-f006:**
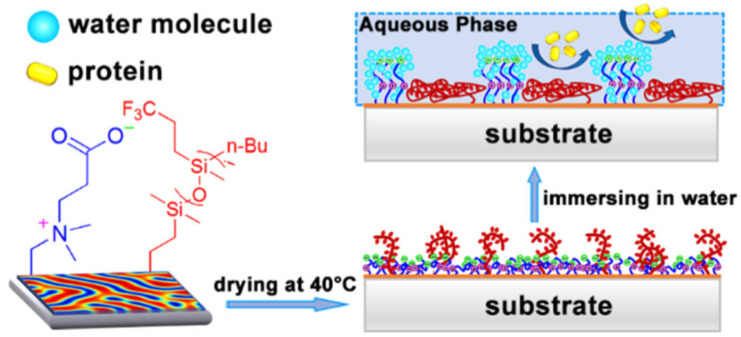
Antifouling mechanism of amphiphilic copolymers [30] with permission (Copyright © 2023, American Chemical Society).

**Figure 7 biomimetics-08-00200-f007:**
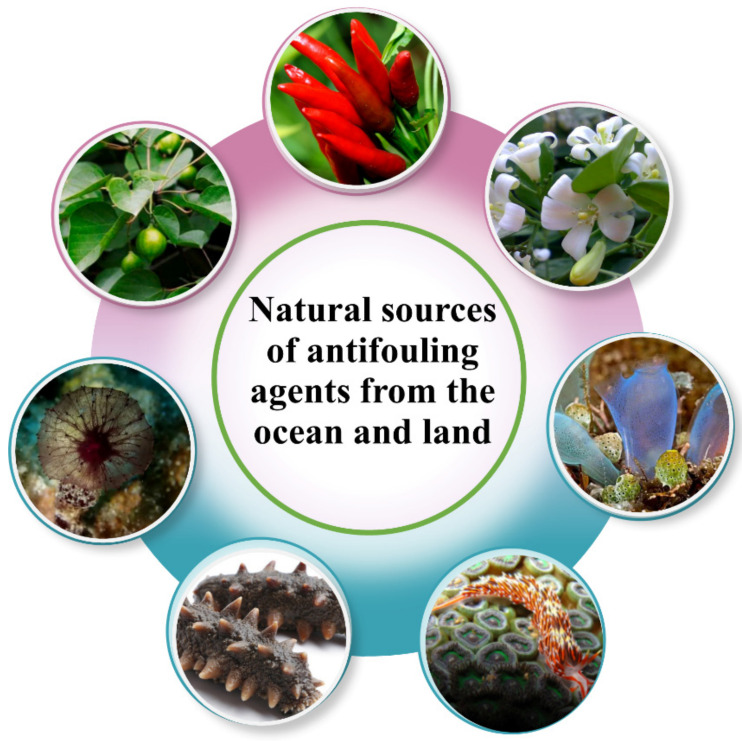
Natural sources of antifouling agents from the ocean and land.

**Figure 8 biomimetics-08-00200-f008:**
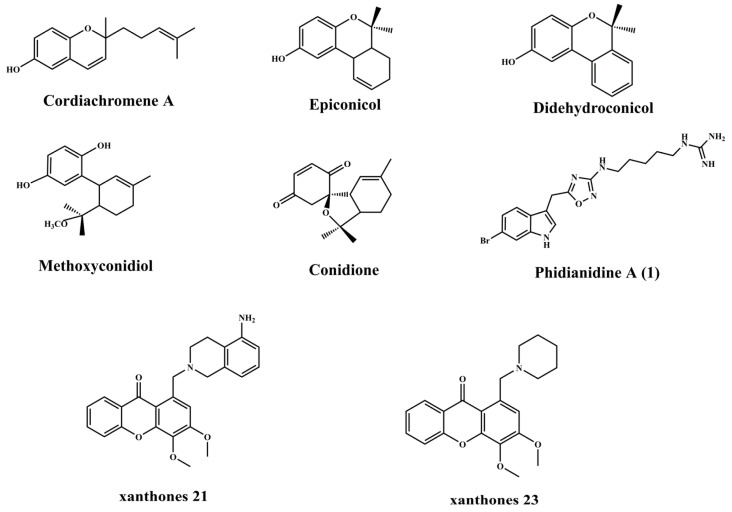
The molecular structure of antifouling agents [35,43,47].

**Figure 9 biomimetics-08-00200-f009:**
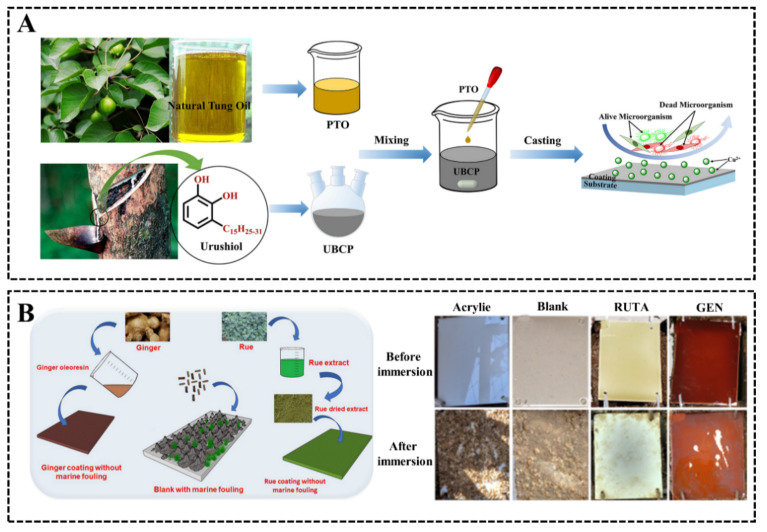
(**A**) Preparation process and antifouling mechanism of UBCP/PTO composite coating [41] with permission (Copyright © 2023, Elsevier). (**B**) Preparation process of silicone resin with rue and ginger oleoresin extract and antifouling results after 6 months of marine field tests [42] with permission (Copyright © 2023, Springer Nature).

**Figure 10 biomimetics-08-00200-f010:**
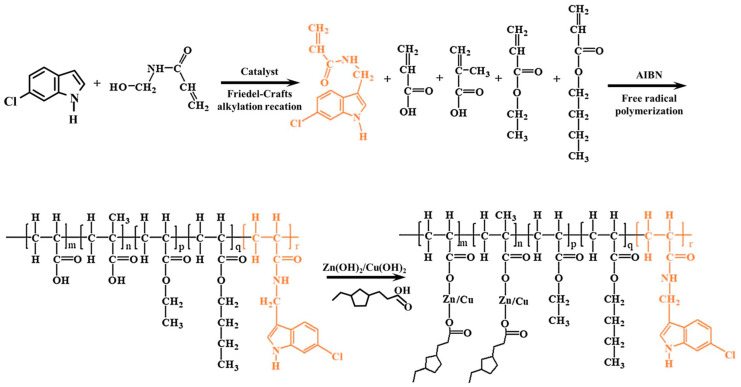
Synthesis of acrylic metal salt resins containing the indole derivative structure [46] with permission (Copyright © 2023, Elsevier).

**Figure 11 biomimetics-08-00200-f011:**
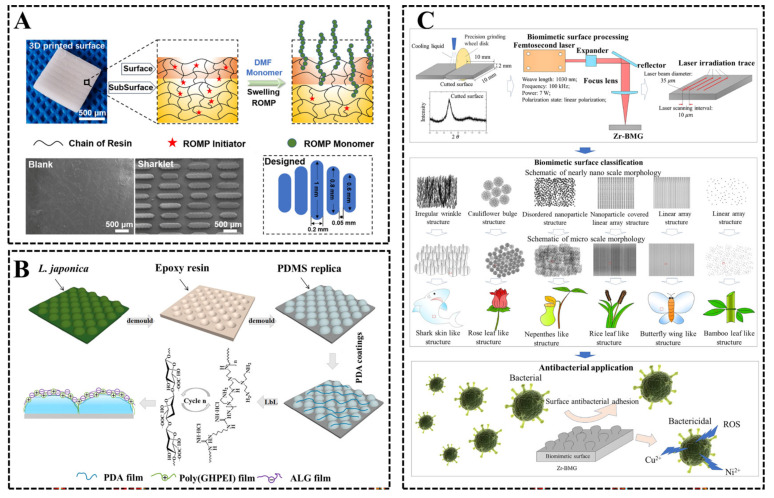
(**A**) Schematic diagram of the preparation of poly (ionic liquid) brush-grafted “Sharklet” surface [55] with permission (Copyright © 2023, Elsevier). (**B**) Schematic diagram of the preparation of alternate deposition of sodium alginate and poly (GHPEI) on PDMS substrates constructed with microstructure morphology imitating the surface of *Laminaria japonica* using a layer-by-layer assembly method [58] with permission (Copyright © 2023, Elsevier). (**C**) Schematic diagram of the mechanism of preparing bionic structures using femtosecond laser [60] with permission (Copyright © 2023, Elsevier).

**Figure 12 biomimetics-08-00200-f012:**
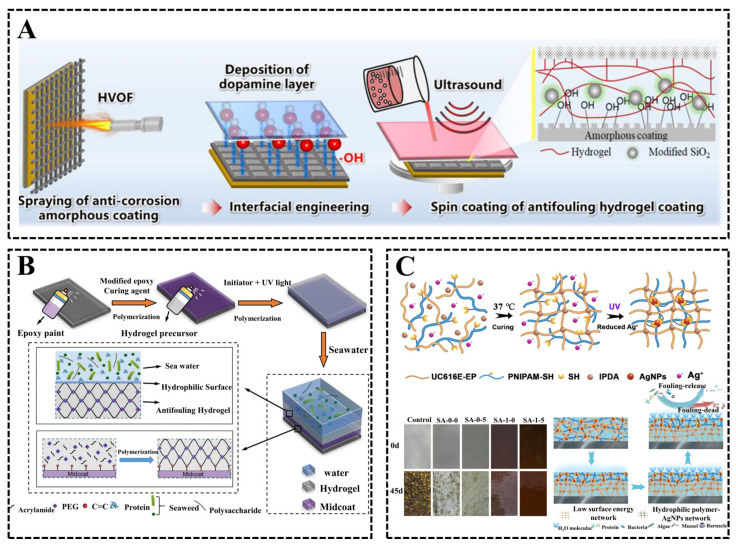
(**A**) Schematic diagram of preparation of hydrogel-anchored iron-based amorphous (HAM) coating [66] with permission (Copyright © 2023, American Chemical Society). (**B**) Illustration of the process of preparing a coatable anti-sewage gel coating based on polyacrylamide-cross-linked dobby polyethylene glycol [67] with permission (Copyright © 2023, John Wiley and Sons). (**C**) Procedure of preparation of bio-based amphiphilic hydrogel coating and antifouling mechanism schematic [68] with permission (Copyright © 2023, Elsevier).

**Figure 13 biomimetics-08-00200-f013:**
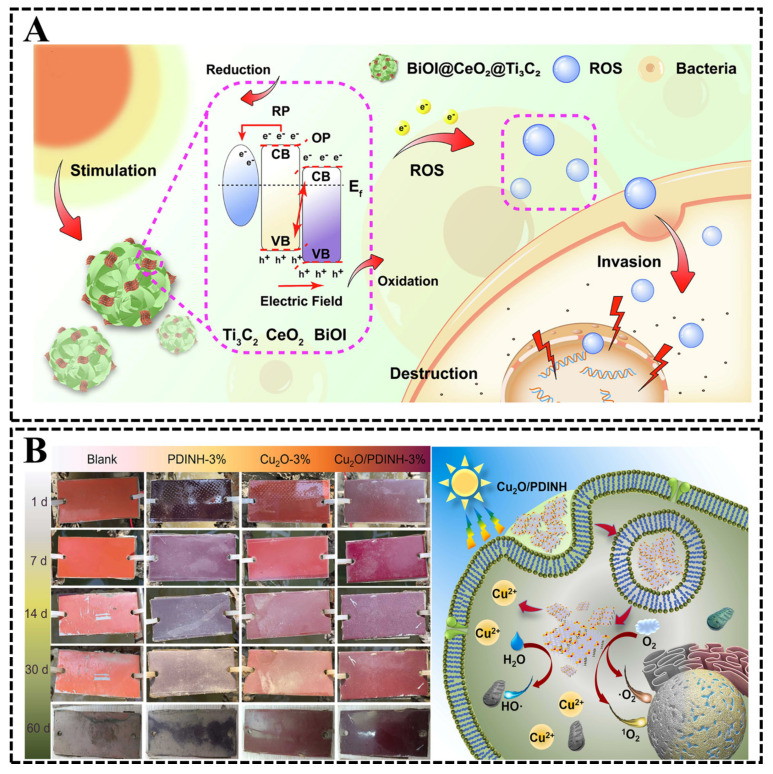
(**A**) Antibacterial mechanism of BiOI@CeO_2_@Ti_3_C_2_ [75] with permission (Copyright © 2023, Elsevier). (**B**) Photos of blank, PDINH3%, Cu_2_O-3% and Cu_2_O/PDINH-3% coatings after immersion in seawater for 1, 7, 14, 30 and 60 days, respectively, and schematic diagram of antibacterial mechanism of Cu_2_O/PDINH-3% coatings [78] with permission (Copyright © 2023, Elsevier).

**Figure 14 biomimetics-08-00200-f014:**
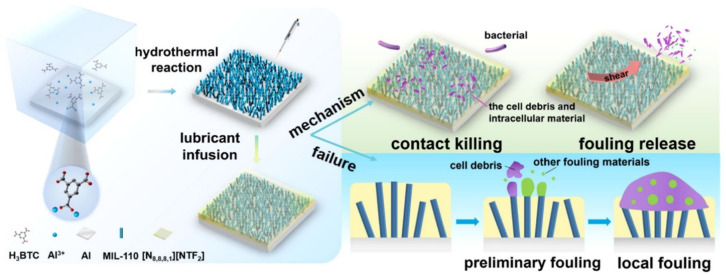
Schematic diagram of the preparation process and antifouling mechanism of SLIPS [83] with permission (Copyright © 2023, Elsevier).

**Figure 15 biomimetics-08-00200-f015:**
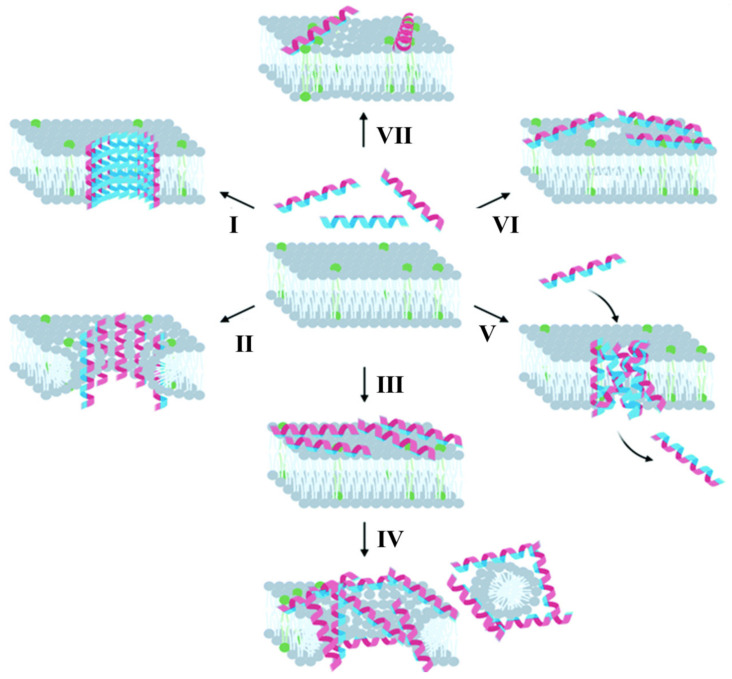
Action mechanism of antimicrobial peptides [87,88] with permission (Copyright © 2023, Royal Society of Chemistry). The membrane is shown in light gray and the negatively charged phospholipids are highlighted in green. The amphipathic nature of the peptide is demonstrated by the dual coloring of the helix: blue represents the surface with positively charged residues and red represents the surface with hydrophobic residues. (I) AMPs are inserted vertically into the phospholipid bilayer in the manner of barrel plates (barrel model). (II) Peptides disrupt the transmembrane potential and osmoregulatory function of cells, inhibit cellular respiration, and eventually lead to the death of the bacterium (ring pore model). (III) Peptides are arranged parallel to the membrane surface like a carpet (carpet model). (IV) When the antimicrobial peptide reaches a certain concentration, it may lead to cell membrane disintegration and extracellular material extravasation (detergent model). (V) Peptides are inserted into cell membranes in the form of aggregates. (VI) The charge of the peptide accumulated on the outer surface of the cell membrane creates a high enough potential difference to cause pores to form in the cell membrane (electroporation model). (VII) Phospholipid clustering can change the morphology of the membrane.

**Figure 16 biomimetics-08-00200-f016:**
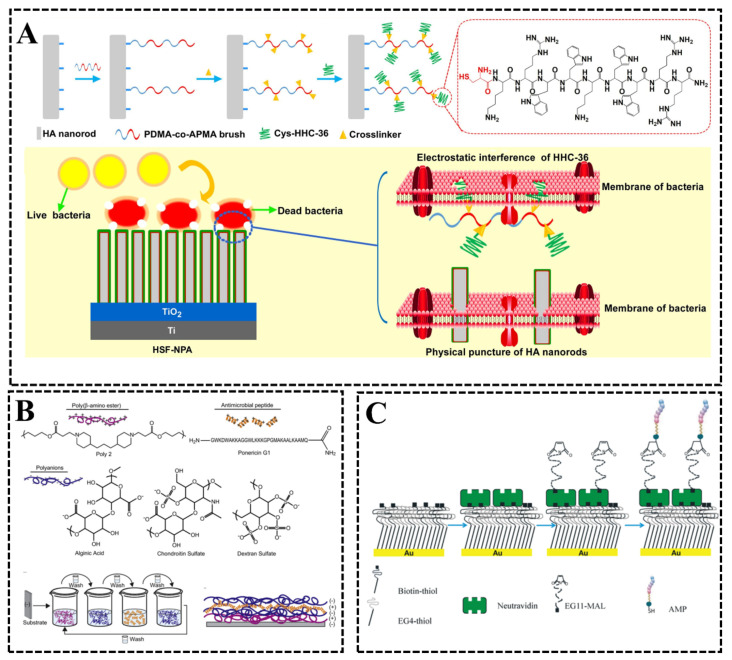
(**A**) Illustration of the preparation process and synergistic bactericidal mechanism of antimicrobial peptide Cys-HHC-36 linked to HA nanorods [92] with permission (Copyright © 2023, Elsevier). (**B**) Molecular structure of each component in the self-assembly process and the preparation process of AMP-doped self-assembled films [93] with permission (Copyright © 2023, Elsevier). (**C**) Schematic diagram of the preparation of antimicrobial peptide MSI-78A immobilized onto gold surface [95] with permission (Copyright © 2023, Springer Nature).

**Table 1 biomimetics-08-00200-t001:** Basic information on different antifouling strategies.

Antifouling Strategies	Mechanisms	Compounds	Advantages and Disadvantages	References
Fouling release coatings	“Baier curve”; low-surface-energy materials make it difficult for contaminated organisms to adhere to the surface	Low-surface-energy silicones and Fluoropolymers; fluoro; silicone co-modified materials	Commercialization poor adhesion on the hull surface; easily damaged	[16,17,18,19,20,21,22,23,24,25,26,27,28,29,30,31]
Natural antifouling agent	Marine and terrestrial plants and animals resist the attachment of fouling organisms by secreting active substances	Steroids, fatty acids, amino acids, indoles, alkaloids and Synthetic analogs	Excellent biocompatibility and degradability; poor stability; difficulty with separation and purification	[32,33,34,35,36,37,38,39,40,41,42,43,44,45,46,47]
Micro/nano structured materials	“Attachment point theory”; the different types of micro/nanostructures on plant and animal surfaces resist the attachment of fouling organisms	Microstructural morphology mimics shark skin, mussel and lotus leaves, etc.; regularized artificial microstructures surface	Low risk to the marine ecological environment; difficult processing; high production cost	[48,49,50,51,52,53,54,55,56,57,58,59,60,61]
Hydrogel coatings	Inspired by the mucus secreted by the epidermis of fish and amphibians to resist attachment of fouling organisms	Amphiphilic Copolymers; polyethylene glycol (PEG); polyacrylamide (PAM), etc.	Enhanced mechanical stability and excellent antifouling activity; need to conduct further sea testing	[62,63,64,65,66,67,68,69]
Photocatalytic materials	Based on semiconductor photocatalyst to decompose seawater and dissolved oxygen to produce reactive oxygen species (ROS) for antifouling	Titanium oxide (TiO_2_), zinc oxide (ZnO) and Cerium dioxide (CeO_2),_ etc.	Restrictions on use in dark conditions	[70,71,72,73,74,75,76,77,78,79]
Slippery liquid-infused porous surfaces (SLIPS)	Inspired by the special microstructure and smooth properties of natural pigweed	Microstructure; lubricants (perfluoro polyethers and silicone oils)	Self-healing; vulnerable; unclear impacts on marine ecosystems	[80,81,82,83,84]
Antimicrobial-peptide-modified surfaces	Antibacterial peptides interact electrostatically with bacterial cell membranes to disrupt membrane integrity	Antimicrobial peptide	Environmentally friendly; excellent antifouling performance; maintaining spatial structure and antimicrobial properties remains a challenge	[85,86,87,88,89,90,91,92,93,94,95,96,97,98,99,100,101]

## Data Availability

The data presented in this study are available on request from the corresponding author.

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
