# Peer review of "Research Progress on New Environmentally Friendly Antifouling Coatings in Marine Settings: A Review"

_biomimetics, 2023, doi:10.3390/biomimetics8020200_

Round 1

Reviewer 1 Report

This Review deals with new environmental friendly antifouling for coatings in marine application. This topic is worthy of interest and I believe that can be useful for readers. 

The Review is well organized and deserves publication after minor changes.

Line 65: reference or references must be added.

Line 67: reference or references must be added.

Line 144: reference or references must be added.

Figure 8. Please structure of Methoxyconidiol must be drawn correctly (OH on the left must be drawn HO; OCH3 must be drawn CH3O).

Lines 298 and 300: authors report "m-phenylenediamine A (1)" what is A? what is (1)? please clarify.

Lines 315, 316, 317: authors discuss about "xanthone 21 and 23". These structures are not reported, please add these structures.

Line 499: TiO2, please write correctly.

Line 508, 511: what is BiOI?

Figure 15 is difficult to follow.

Author Response

Response to Reviewer 1 Comments

To Reviewer 1: Thanks for the reviewers’ valuable comments. The revised content has been marked in red in the revised manuscript.

Point 1: Line 65: reference or references must be added.

Response 1: Thanks for the reviewers’ comments. Relevant references are given in the manuscript.

Point 2. Line 67: reference or references must be added.

Response 2: Thanks for the reviewers’ comments. Relevant references are given in the manuscript.

Point 3. Line 144: reference or references must be added.

Response 3: Thanks for the reviewers’ comments. Relevant references are given in the manuscript.

Point 4. Figure 8. Please structure of Methoxyconidiol must be drawn correctly (OH on the left must be drawn HO; OCH3 must be drawn CH3O).

Response 4: Thanks for the reviewers’ comments. The relevant molecular structures in the manuscript have been corrected.

Point 5. Lines 298 and 300: authors report "m-phenylenediamine A (1)" what is A? what is (1)? please clarify.

Response 5: Thanks for the reviewers’ valuable comments. I am sorry for the trouble caused to the reviewers, as the author's carelessness did not catch this error when writing. Here m-phenylenediamine A (1) should be Phidianidine A (1), which is the name of a marine natural antifouling active substance, where A and (1) have no special designation. the molecular structure of Phidianidine A (1) is given in the manuscript.

Point 6. Lines 315, 316, 317: authors discuss about "xanthone 21 and 23". These structures are not reported, please add these structures.

Response 6: Thanks for the reviewers’ comments. We have added the molecular structure of the relevant antifouling agent to the manuscript.

Point 7. Line 499: TiO2, please write correctly.

Response 7: Thanks for the reviewers’ comments. The author has corrected this error in the manuscript.

Point 8. Line 508, 511: what is BiOI?

Response 8: Thanks for the reviewers’ comments. BiOI is a novel semiconductor material in ternary oxyhalide BiOX (X = Cl, Br, and I) with excellent optical properties, high chemical stability, non-toxicity, corrosion resistance and other favorable characteristics. Moreover, BiOI can be used to modify wide bandgap semiconductor materials to improve photocatalytic efficiency by forming heterojunctions. Relevant content has been added to the manuscript.

Point 9. Figure 15 is difficult to follow.

Response 9: Thanks for the reviewers’ valuable comments. We are very sorry for the trouble this figure caused the reviewers, and we cite it in the reference "Gan, B.H.; Gaynord, J.; Rowe, S.M.; Deingruber, T.; Spring, D.R. The multifaceted nature of antimicrobial peptides: current synthetic chemistry approaches and future directions. Chem. Soc. Rev. 2021, 50, 7820-7880."

The figure explains the antimicrobial peptide Several mechanisms of action are schematically explained, with the helical strip model representing the antimicrobial peptide and the rectangular shape model representing the phospholipid bilayer. For example, in I, the antimicrobial peptide is inserted vertically into the phospholipid bilayer like a barrel plate, so it is called the "barrel" model, and in III, the AMPs are arranged parallel to the cell membrane surface like a carpet, so it is called the "carpet" model. The other serial numbers indicate an interaction model. The diagram shows the different modes of interaction between antimicrobial peptides and phospholipid bilayers. For the sake of understanding, we have added the meaning of the colors in the figure, which is given in the figure name. "The membrane is shown in light gray and the negatively charged phospholipids are highlighted in green. The amphipathic nature of the peptide is demonstrated by the dual coloring of the helix: blue represents the surface with positively charged residues and red represents the surface with hydrophobic residues."

I believe that the above answer can solve the reviewer's confusion about this figure.

Reviewer 2 Report

Overall the review is well written with all the context. A few minor edits can be done to improve the readability of the review like

- improving the transition from the description of one paper to another

- why anti-microbial peptide anti-fouling method is a separate section

- having a table for different anti-fouling methods and their advantages will improve the readability

A major concern that I have is about the significant contribution. How is this review different than the previously published review?

"Gu, Y.Q.; Yu, L.Z.; Mou, J.G.; Wu, D.H.; Xu, M.S.; Zhou, P.J.; Ren, Y. Research strategies to develop environmentally friendly marine 785 antifouling coatings. Mar. Drugs 2020, 18, 371. "

The review is well-written and the quality of the English is acceptable.

Author Response

Response to Reviewer 2 Comments

To Reviewer 2: Thanks for the reviewers’ valuable comments. The revised content has been marked in green in the revised manuscript.

Point 1: improving the transition from the description of one paper to another

Response 1: Thanks for the reviewers’ comments. The authors read the manuscript carefully, improving and adding descriptions between transitions from one paper to another, and the changes are shown in green in the manuscript.

Point 2: why anti-microbial peptide anti-fouling method is a separate section

Response 2: Thanks for the reviewers’ comments. Because our group is mainly engaged in research on the preparation of antifouling surfaces based on antimicrobial peptide modifications. The antimicrobial peptide is a biomolecule secreted by many organisms that can inhibit the adhesion of fouling organisms such as bacteria and have excellent cytocompatibility and high affinity, which are potential materials for the preparation of environmentally friendly antifouling coatings. In recent years, they have been favored by many researchers. Therefore, we separate the antifouling means of antimicrobial peptides and discuss the source of antimicrobial peptides, the antimicrobial mechanism and the preparation means of antimicrobial peptide-modified surfaces as a reference for the research of environmentally friendly antifouling coatings.

Point 3: having a table for different anti-fouling methods and their advantages will improve the readability

Response 3: Thanks for the reviewers’ comments. The authors have given relevant tables in the manuscript (Table 1), which mainly discuss the basic information about different antifouling strategies and their respective advantages and disadvantages.

Table 1. Basic information on different antifouling strategies

Antifouling

strategies

Mechanisms

Compounds

Advantages and

disadvantages

References

Fouling releases coatings

"Baier curve";

low surface energy materials make it difficult for contaminated organisms to adhere to the surface

Low surface energy silicones and fluoropolymersï¼›fluoro; silicone co-modified materials

Commercialization

poor adhesion on the hull surface; easily damaged

[19-31]

Natural

antifouling agent

Marine and terrestrial plants and animals resist the attachment of fouling organisms by secreting active substances

Steroids, fatty acids, amino acids, indoles, alkaloids, and Synthetic analogs

Excellent

biocompatibility and degradability; poor stability; difficulty with separation and purification

[34-47]

Micro/nano

structured

materials

"Attachment point theory";

the different types of micro-nano structures on plant and animal surfaces resist the attachment of fouling organisms

Microstructural

morphology mimics shark skin, mussel and lotus leaves, etc.

regularized artificial microstructures

surface

Low risk to the marine ecological environment;

difficult processing;

high production cost

[51-61]

Hydrogel

coatings

Inspired by the mucus secreted by the epidermis of fish and amphibians to resist attachment of fouling organisms

Amphiphilic

Copolymers;

polyethylene glycol (PEG);

polyacrylamide (PAM), etc.

Enhanced mechanical

stability and excellent antifouling activity;

need to conduct further sea testing

[62-69]

Photocatalytic materials

Based on semiconductor photocatalyst to decompose seawater and dissolved oxygen to produce reactive oxygen species (ROS) for antifouling

Titanium oxide (TiO2), zinc oxide (ZnO), and Cerium dioxide (CeO2), etc.

Restrictions on use in dark conditions

[72-79]

Slippery liquid-infused porous surfaces (SLIPS)

Inspired by the special microstructure and smooth properties of natural pigweed

Microstructure;

lubricants (perfluoro polyethers and

silicone oils)

Self-healing;

vulnerable; unclear

impacts on marine

ecosystems

[80-84]

Antimicrobial peptide-modified surfaces

Antibacterial peptides interact electrostatically with bacterial cell membranes to disrupt membrane integrity

Antimicrobial

peptide

Environmentally-friendly;excellent antifouling performance;

maintaining spatial structure and

antimicrobial

properties remains a challenge

[89-101]

Point 4: A major concern that I have is about the significant contribution. How is this review different than the previously published review?

"Gu, Y.Q.; Yu, L.Z.; Mou, J.G.; Wu, D.H.; Xu, M.S.; Zhou, P.J.; Ren, Y. Research strategies to develop environmentally friendly marine antifouling coatings. Mar. Drugs 2020, 18, 371. "

Response 4: Thanks for the reviewers’ comments. The authors have carefully read the references proposed by the reviewers and concluded that there are many differences between this manuscript and the references proposed by the reviewers. In this manuscript, we have re-summarized the recent research advances in environmentally friendly antifouling coatings, such as fouling-release antifouling coatings, natural antifouling agent materials, micro/nano-structured antifouling materials, hydrogel antifouling materials, photocatalytic antifouling materials, SLIPS antifouling materials, and antifouling materials based on antimicrobial peptide modification. Compared with the references proposed by the reviewers, we have added a review of the latest research progress on hydrogel antifouling materials, SLIPS antifouling materials and antimicrobial peptide-based modified antifouling materials. Other differences are as follows.

(1) In fouling-release antifouling coatings we discuss the most widely used substrate material polydimethylsiloxane (PDMS) elastomers, and then focus on the means used in recent years to improve the performance of fouling-release antifouling coatings based on their performance deficiencies, such as by doping PDMS with nanoparticles and grafting functionalities such as amphoteric, PEG and quaternary ammonium salt groups or the preparation of fluoro silicone co-modified materials.

(2) In the section on natural antifouling agents we discuss the application of natural active substances derived from terrestrial plants and marine flora and fauna in antifouling coatings and the progress of the application of synthetic strategies to form similar antifouling active substances, using LC50, EC50 and LC50/EC50 values to evaluate the antifouling activity of natural active substances. The construction of antifouling coatings doped with natural antifouling agent components based on synergistic strategies is further discussed.

(3) In micro/nano-structured antifouling materials, we discuss the progress of research on surface microstructures of mimicking marine and terrestrial plants and animals, where the mimicking sources of surface microstructures of plants and animals in the ocean are sharks, mussels and kelp. Terrestrial flora and fauna such as cicadas rose leaves and rice leaves. The preparation and antifouling properties of regularized artificial surface microstructures are further discussed.

(4) In photocatalytic antifouling coatings we first focus on the mechanism of action of photocatalytic antifouling. Then, based on the performance defects of currently used photocatalytic coatings, we discuss the preparation methods used to improve the performance of photocatalytic coatings, and their respective advantages and disadvantages.

(5) Based on the progress of our group in antifouling research on antimicrobial peptides, we focus on the sources of antimicrobial peptides, antimicrobial mechanisms and means of preparation of antimicrobial peptide-modified surfaces to provide a reference for research on environmentally friendly anti-fouling coatings.

Of course, there are still a large number of differences in the manuscript, and probably there are some similarities between this manuscript and the framework of the references proposed by the reviewer, but the discussions in this manuscript are all different from the above, and we are very sorry for confusing the reviewer, and the authors believe that the above explanation can answer this question well.

Round 2

Reviewer 2 Report

I am satisfied with the author's revision.

Moderate editing is required to improve the readability